# Radiative Transfer Model Comparison with Satellite Observations over CEOS Calibration Site Libya-4

Yves Govaerts* 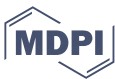, Yvan Nollet and Vincent Leroy

Rayference, 1030 Brussels, Belgium
* Correspondence: yves.govaerts@rayference.eu

**Abstract:** Radiative transfer models of the Earth's atmosphere play a critical role in supporting Earth Observation applications such as vicarious calibration. In the solar reflective spectral domain, these models usually account for the scattering and absorption processes in the atmosphere and the underlying surface as well as the radiative coupling between these two media. A range of models is available to the scientific community with built-in capabilities making them easy to operate by a large number of users. These models are usually benchmarked in idealised but often unrealistic conditions such as monochromatic radiation reflected by a Lambertian surface. Four different 1D radiative transfer models are compared in actual usage conditions corresponding to the simulation of satellite observations. Observations acquired by six different space-borne radiometers over the pseudo-invariant calibration site Libya-4 are used to define these conditions. The differences between the models typically vary between 0.5 and 3.5% depending on the spectral region and the shape of the sensor spectral response.

**Keywords:** radiative transfer model; benchmarking; vicarious calibration

## 1. Introduction

Radiative transfer models (RTMs) are extensively used in Earth Observation (EO) to support applications such as sensitivity analysis for the preparation of new space missions [1], retrieval algorithm verification [2], look-up table generation [3], satellite data assimilation [4] or radiometric calibration verification [5]. RTMs simulate electromagnetic radiation propagation in highly idealised media, which is typically a 1D plane-parallel atmosphere with an underlying flat surface. A large scientific community with a scattered background in radiation transfer theory operates these models. Typical RTMs implement one or more numerical methods to solve the radiative transfer equation representing radiation transport in homogeneous layers. The vertical structure of the atmosphere is described as a stack of uniform plane-parallel layers, possibly with different optical properties. Some of these models (e.g., [6–9]) leave to the user the task of specifying the radiative properties of each atmospheric layer. Such an approach might become cumbersome when many simulations must be performed and requires advanced knowledge in atmospheric radiative properties. To ease RTM usage, many of these models include pre-processing functionalities that calculate the mean optical properties of each uniform layer accounting for possible radiative processes such as scattering, absorption or emission by molecules, dry matter, water or ice. These user-friendly features allow the users to perform simulations specifying only a limited number of parameters such as total gas concentration or aerosol optical thickness.

These models often implement six standard atmospheric vertical profiles representing typical states of the atmosphere at different seasons and locations [10]. These profiles describe the atmospheric vertical composition from 0 to 120 km providing the volume mixing ratio for 28 molecules. They have been initially implemented in the LOWTRAN (Low-Resolution Atmospheric Transmittance and Radiance) model [11]. Although these profiles have been included in most atmospheric radiative transfer models such as MODTRAN (Moderate Resolution Atmospheric Transmittance and Radiance) [12], Libradtran

(Libraries of Radiation Transfer) [13] or 6SV (Second Simulation of a Satellite Signal in the Solar Spectrum) [14], differences exist in their implementation, such as the actual number of radiatively active molecules or the number of horizontal layers. Some models allow one to rescale the total column concentration of some species included in these standard profiles, which is a convenient way to account for the high spatial and temporal variability of water vapor concentration. A similar feature is also needed to take into account the constant greenhouse gases concentration increase such as carbon dioxide ($CO_2$) or methane ($CH_4$). The estimation of the Rayleigh optical thickness also relies on these profiles.

The radiative transfer equation is strictly valid for monochromatic electromagnetic radiation, i.e., composed of a single frequency, which seriously complexifies the modelling of satellite observations. Simulations of radiances acquired by a radiometer such as MODIS (Moderate Resolution Imaging Spectroradiometer) require to integrate the simulated radiative field over the observed spectral domain. The most obvious way to perform this spectral integral consists of solving the radiative transfer equation for each wavelength of a spectral mesh dense enough to resolve the line structure of the absorption spectrum of the atmosphere. This so-called "line-by-line" approach requires calculating the contribution of each spectral line for all participating molecules in all the atmospheric layers. The computational time might quickly become prohibitive in the case of large spectral bands. A faster but less accurate approach considers the absorption in a spectral interval characterised by a set of pre-calculated coefficients that depend on temperature and other parameters such as the pressure. There is no unique method (e.g., [15,16]) to approximate transmittance in a spectral interval which might also lead to systematic differences in the calculated radiation field among models that include molecular absorption estimation.

It is therefore critical to thoroughly benchmark RTMs to quantify the impact of these different numerical approximations on the accuracy of the modelled satellite observation. Standard RTM comparison initiatives typically rely on a set of ideal cases, e.g., monochromatic radiation or Lambertian surface [17,18] that are not necessarily representative of how RTMs are used in practice in the context of EO applications [19]. These authors compared two radiative transfer models, 6SV and MODTRAN, in the context of satellite image atmospheric correction. They reported differences as large as 11% between these two models. Each of these RTMs implement different parametrisation for, e.g., Rayleigh scattering, molecular absorption, number of active molecules or number of quadrature points. The numerical method implemented to solve the radiative transfer equation also differs. Discrepancies of a few percent between RTMs have also been reported by Escribano et al. [20] and Vicent et al. [3] who compared different models for data assimilation and look-up table generation, respectively.

This paper compares four RTMs in actual usage conditions as those employed by vicarious calibration methods. The selected case consists of simulated radiance over a bright pseudo-invariant calibration site (PICS) [21]. Such an application requires the simulation of top-of-atmosphere Bidirectional Reflectance Factor (BRF) in the 400–2500 nm spectral interval. These models are used in their standard configuration as delivered to end-users. Only minor modifications have been applied to some models to allow the definition of exactly the same surface conditions. This goal is motivated by the constant radiometric improvement of new EO missions with uncertainties as low as 3% or even lower [5].

To achieve this objective, satellite observations acquired by six different radiometers over Libya-4, a bright desert PICS, have been simulated with four different RTMs. This paper focuses on quantifying the discrepancies between these RTMs operated with similar input parameters. The parameters characterising the radiative properties of Libya-4 are described in Section 2. Section 3 briefly summarises the four RTMs. Simulated satellite data are listed in Section 4. Finally, the comparison method is explained in Section 5, and the results are presented in Section 6.

## 2. Libya-4 Site Characterisation

The Libya-4 desert area, located in the Great Sand Sea, is centered around 28.55° N and 23.39° E with a maximum spatial extension of about $1° \times 1°$. It is one of the most essential bright desert PICS recommended by the Committee on Earth Observation Satellites (CEOS) for its size and radiometric stability. Libya-4 morphology consists of oriented sand dunes shaped by dominant trade winds. These dunes are organised at multiple spatial scales with a mean altitude of about 120 m. The central part of this domain consists of large-scale north–south transverse dunes, i.e., ridges of sand with a steep face on the downwind side [22]. The northeast part of the area exhibits lower altitude with crescent dune type [23]. Among the CEOS bright desert PICS, Libya-4 is one of the most stable and widely used for radiometer stability monitoring, inter-calibration or absolute calibration determination [24].

The possibility to accurately simulate satellite observations acquired in the visible, near infra-red and shortwave infra-red regions over Libya-4 has already been demonstrated in previous studies. It is possible to simulate observations acquired over Libya-4 with a mean accuracy of a few percent when several tens of clear-sky data are considered [21,25–27]. These results rely on the characterisation of the following surface and atmosphere properties:

- The surface BRF is characterised by the so-called RPV (Rahman–Pinty–Verstraete) model [28]. The values of the RPV parameters have been derived at 1 nm resolution with the method described in [25], including the improvements proposed by [21]. The 4-parameter version of the RPV model is used for this study [29].
- Mid-latitude summer atmospheric vertical profile [10] is assumed all year long. Water vapor and ozone total column concentrations are rescaled with the values taken from the European Centre for Medium-Range Weather Forecasts (ECMWF) ERA5 dataset [30].
- A specific aerosol optical thickness climatology assuming non-spherical particles representing Saharan dust is used as described in [21].

## 3. Description of Selected RTMs

### 3.1. 6sv Version 1.1

6SV is an advanced RTM released by the MODIS Land Surface Reflectance Science Computing Facility. It simulates the reflection of solar radiation by a coupled atmosphere–surface system for a wide range of atmospheric, spectral, and geometric conditions [14]. The 6SV code is based on the method of successive orders of scattering (SOS). It accounts for the polarisation of radiation in the atmosphere through the calculation of the Q and U components of the Stokes vector, although this feature has not been used in this study. The intensity is successively computed for photons scattered one, two, three times, etc. with the total intensity obtained as the sum of all orders. Numerical integration is calculated using the decomposition in Fourier series for the azimuth angle and Gaussian quadrature for the zenith angle. The code has been compiled with the default configuration, i.e., the atmosphere is divided into 30 horizontal layers, the number of zenith (azimuth) calculation angles is set to 25 (181), and 83 Legendre coefficients are used to characterise the aerosol phase function. The routines dealing with the RPV model have been modified to support the 4-parameter version [29].

Molecular absorption is pre-computed from the HITRAN96 database [31] at a resolution of 10 cm$^{-1}$ [32]. In this approach, two random exponential band models are selected for water vapor [33] and all other gases [34], respectively. When several gases exhibit absorption bands in the same spectral interval, the total transmission is set equal to the product of each gas transmission. The following six gases are contributing to the molecular absorption: water vapor ($H_2O$), ozone ($O_3$), oxygen ($O_2$), carbon dioxide ($CO_2$), methane ($CH_4$), nitrous oxide ($N_2O$) and carbon monoxide (CO). The six standard atmospheric profiles proposed by [10] are included in the code, allowing for the rescaling of water vapor and ozone concentration. The absorption by atmospheric gases is computed separately as a simple multiplicative factor [35]. Coupling between aerosol scattering and gaseous

transmittance is thus correctly calculated for the first order of scattering but approximated or neglected for the other orders of scattering. The applied approximation is valid for spectral regions where the molecular transmittance is high, i.e., larger than 0.98. 6SV can thus simulate the radiance field propagating through the atmosphere with good accuracy in spectral regions where the coupling between molecular absorption and scattering processes can be neglected. It has limited accuracy in spectral regions where both effects are present [32].

6SV has been extensively compared with other RTMs such as RT3 (Radiative Transfer) [36], MODTRAN4 [37], and SHARM (Spherical Harmonics) [6] under different types of atmosphere [18]. These authors concluded that RT3 demonstrates better agreement with 6SV than MODTRAN4 which does not include polarisation. Its average difference with 6SV varies within 0.5–3.0% for all considered cases.

### 3.2. Rtmom Version 1

RTMOM (Radiative Transfer Matrix Operator Method) model was originally designed and developed at EUMETSAT to support the operational vicarious calibration of Meteosat satellites [38]. It implements the matrix operator method [39,40] to solve the radiative transfer equation [41]. The main advantage of this method concerns the simulation of light propagation in optically dense media. Unlike the SOS method, the computational time is independent of the optical thickness. The code calculates azimuthally resolved radiances at a discrete number of incident and observation zenith angles or quadrature points at pre-defined horizontal levels. Reflection, transmission and source operators for each layer are obtained by repeated application of the doubling algorithm to an optically very thin sub-layer for which the single scattering approximation holds. The individual layers are then combined using the adding algorithm. The final output consists of the diffuse light field at the layer boundaries for all selected solar incident angles.

The atmosphere is discretised with 50 horizontal layers in which the radiative transfer equation is solved with 16 angular quadrature points. The aerosol phase function is represented with 31 Legendre coefficients. The delta-fit method is used to cope with the strong forward peak of the phase function that might occur at short wavelengths [42]. Gas transmittance in the spectral interval is approximated with the correlated K-distribution or correlated-k method at 1 nm spectral resolution, which has been generated from the HITRAN96 database for seven molecules ($H_2O$, $O_3$, $O_2$, $CO_2$, $CH_4$, $N_2O$ and CO) for a set of temperature and pressure values. An important operation in the k-distribution method is to sort the spectral grid such that the gas absorption coefficients on the reordered spectral grid is monotonic; with the reordered grid, the absorption spectrum is smooth, and it can be integrated numerically at a very cheap cost using. e.g., a quadrature rule. RTMOM includes the six standard atmospheric profiles proposed by [10] with the possibility to rescale $H_2O$ and $O_3$ total column concentration. Within each atmospheric layer, aerosols, Rayleigh and gas optical thickness are combined to determine the layer total optical thickness. The coupling between scattering and absorption is thus taken into account for any order of scattering.

### 3.3. Libradtran Version 2

libRadtran is a suite of tools for radiative transfer calculations in the Earth's atmosphere which offers several 1D radiative transfer equation solvers such as DISORT (Discrete Ordinate Radiative Transfer Model) [43] or MYSTIC (Monte Carlo code for the phYsically correct Tracing of photons In Cloudy atmospheres) [44,45]. The latter has been used for the current simulations firing 50,000 rays for the simulation of each satellite observation. As for the 6SV code, the RPV surface BRF model has been modified to handle the four-input parameter version of the model.

Within libRadtran V2, the gaseous transmission relies on representative wavelengths parametrisation (REPTRAN) [16]. This method assumes that spectrally integrated radiances can be approximated by weighted means of radiances at so-called representative

frequencies or wavelengths. The representative wavelengths together with their weights are selected by an optimisation method that minimises the deviation from the accurate spectrally integrated radiances for a set of highly variable atmospheric states. These authors reported a REPTRAN maximum inaccuracy of 1%. The transmittance of 14 molecules ($O_3$, $O_2$, $H_2O$, $CO_2$, $NO_2$, BRO, OCLO, HCHO, $O_2$, $SO_2$, $CH_4$, $N_2O$, CO and $N_2$) is accounted for, and their actual concentration can be customised. $CH_4$ and $CO_2$ concentration can thus be set to current values which is essential for the simulations in the shortwave infrared spectral region. The atmosphere is discretised with 50 horizontal layers. The six standard atmospheric profiles proposed by [10] are also included in libRadtran.

*3.4. Artdeco Version 1.1*

ARTDECO (Atmospheric Radiative Transfer Database for Earth Climate Observation) implements three 1D RTE solvers (vector Monte Carlo ray tracing, DISORT and adding–doubling approximation) alongside an input data library to compute Earth atmosphere radiances and radiative fluxes as observed by passive sensors from ultraviolet to thermal infra red (IR) wavelengths. Current simulations are performed with the adding–doubling method for 24 quadrature points. Gas transmittance are represented with the correlated k-distribution method at 10 cm$^{-1}$ spectral resolution and accounts for 10 molecules ($H_2O$, $O_3$, $O_2$, $CO_2$, $CH_4$, $N_2O$, $NO_2$ CO, $SO_2$ and $N_2$). Continuum absorption is applied for the following species: $H_2O$, $O_3$, $CO_2$ and $N_2$. Each gas concentration can be customised. The atmosphere is discretised with 50 horizontal layers. In spectral regions where absorption by many different molecules takes place, the ARTDECO code is extremely slow, i.e., several hours of CPU time for the simulation of one observation.

## 4. Description of the Satellite Data

The study relies on the simulation of actual satellite observations to maximise the realism of the proposed benchmark scenarios. The simulated spectral bands are listed in Table 1 and cover the blue, green, red, near IR (NIR) and shortwave IR (SWIR) spectral regions. These radiometers have a very low radiometric noise. Only clear-sky satellite observations are considered, which means that all pixels over the PICS must be flagged as cloud-free to consider the observation as valid. Each satellite observation in the spectral band $\lambda_k$ has been processed to represent the top-of-atmosphere bidirectional reflectance factor (BRF)

$$\tilde{O}_{\lambda_k}(\Omega, t) = \frac{\pi \tilde{L}_{\lambda_k}(\Omega, t) \, d_\odot^2}{\cos \theta_s(t) \, \tilde{E}_{\lambda_k}} \tag{1}$$

where $\Omega$ represents the geometry of illumination and observation at the acquisition time $t$. $\tilde{L}_{\lambda_k}(\Omega, t)$ is the radiance observed by the sensor in band $\lambda_k$ and $\tilde{E}_{\lambda_k}$ is the exo-atmospheric irradiance in the same spectral band. $d_\odot$ represents the Sun–Earth distance expressed in astronomical units. One year of clear sky observations acquired over Libya-4 have been processed with Equation (1), the illumination and viewing geometries of which are shown in Figure 1. The accuracy of these RTMs to simulate the observations $\tilde{O}_{\lambda_k}(\Omega, t)$ has been assessed in a previous study [27] and is in the range of a few percent.

**Table 1.** List of radiometers used for the RTM comparison scenario. $N_t$ is the number of clear-sky observations.

| Platform | Radiometer | $N_t$ | Bands | | | | | |
| | | | Blue 0.46 μm | Green 0.55 μm | Red 0.65 μm | NIR 0.86 μm | SWIR 1 1.60 μm | SWIR 2 2.20 μm |
|---|---|---|---|---|---|---|---|---|
| AQUA | MODIS | 190 | B03 | B04 | B01 | B02 | B06 | B07 |
| Envisat | AATSR | 150 | – | B1 | B2 | B3 | B4 | – |
| Landsat-8 | OLI | 52 | B1,B2 | B3 | B4 | B5 | B6 | B7 |
| Sentinel-2A | MSI | 58 | B1,B2 | B3 | B4 | B8,B8A | B11 | B12 |
| Envisat | MERIS | 102 | B2,B3 | B5 | B7,B8 | B13,B14 | – | – |
| Sentinel-3A | OLCI | 54 | Oa3,Oa4 | Os6 | Oa7-10 | Oa17 | – | – |

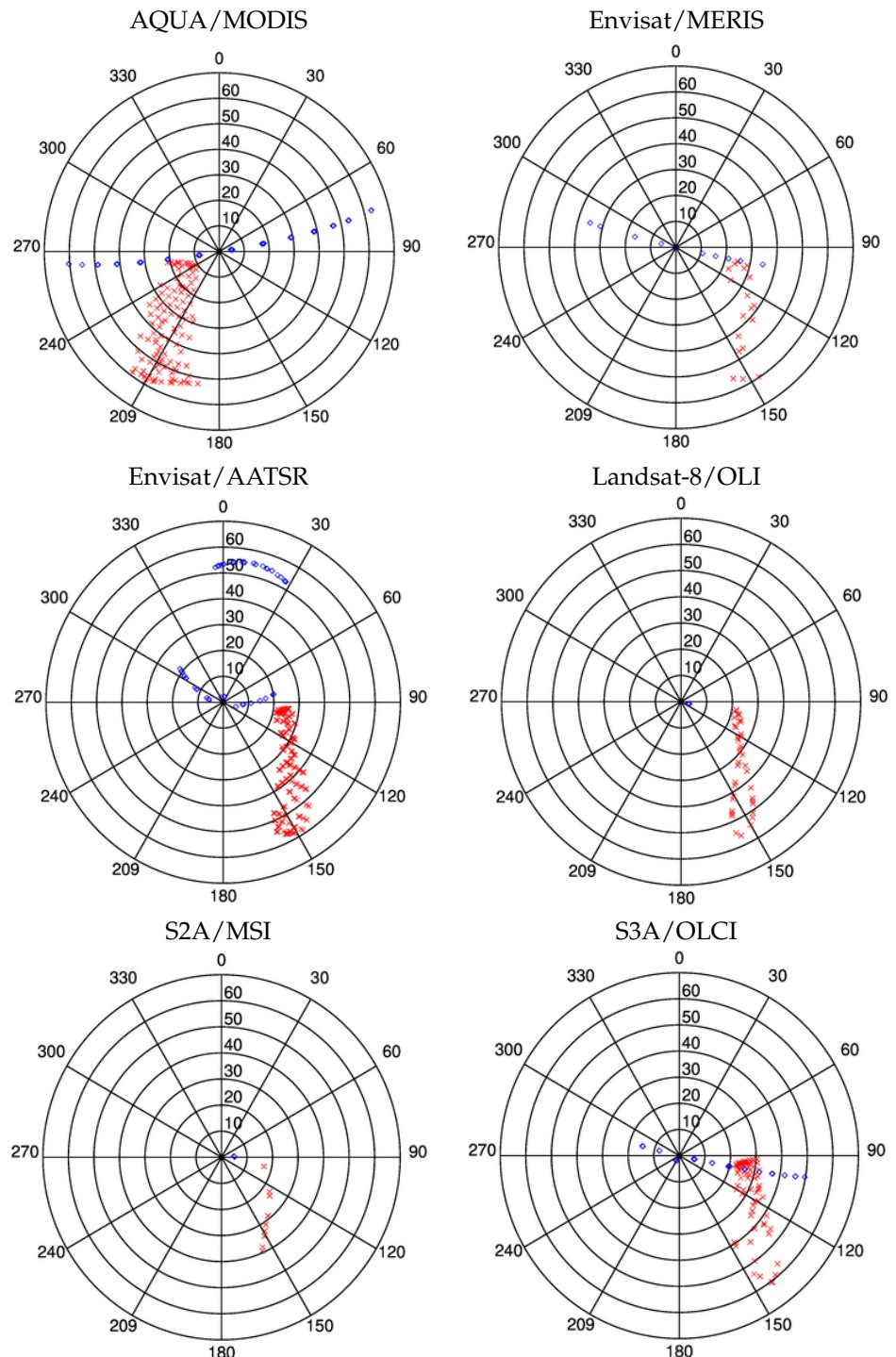

**Figure 1.** Polar plots of the geometry of observation (blue) and illumination (red) for each radiometer. Circles represent zenith angles and polar angles represent azimuth angles with zero azimuth pointing to the North.

## 5. Rtm Comparison Method

Each satellite observation $\tilde{O}_{\lambda_k}(\Omega, t)$ calculated with Equation (1) is simulated with

$$\tilde{R}_{\lambda_k}(\Omega, t; q_i) = \frac{\pi \int_\lambda L(\lambda, \Omega, t; q_i) \xi_k(\lambda) d\lambda \, d_\odot^2}{\cos\theta_s(t) \int_\lambda E(\lambda) \xi_k(\lambda) d\lambda} \tag{2}$$

where $L(\lambda, \Omega, t; q_i)$ is the simulated TOA spectral radiance, $\zeta_k(\lambda)$ is the sensor spectral response of band $k$ of the simulated instrument, and $q_i$ is the set of parameters describing the optical properties of the surface and atmosphere (see Section 2). The Mean Relative Bias (MRB) between the satellite TOA BRFs $\tilde{O}_{\lambda_k}(\Omega, t; q_j)$ and the corresponding simulations $\check{R}_{\lambda_k}(\Omega, t; q_j)$ is expressed as

$$\bar{B}_k = \frac{1}{N_t} \sum_{N_t} \frac{\left(\tilde{O}_{\lambda_k}(\Omega, t) - \tilde{R}_{\lambda_k}(\Omega, t; q_j)\right)}{\tilde{R}_{\lambda_k}(\Omega, t; q_j)} \qquad (3)$$

with $N_t$ being the number of clear-sky observations acquired over Libya-4 for a particular radiometer (Table 1). The MRB $\bar{B}_k$ metric is often used to verify or monitor radiometric calibration over PICS [5,25]. Its dependency on the selected RTM is therefore critical to assess.

Let $m$ with $m \in \{1 \cdots 4\}$ be the RTM index number used to simulate the TOA BRF in Equation (2). $\bar{B}_k(m)$ expresses thus the MRB estimated with RTM $m$. The range of $\bar{B}_k$ values obtained with $M = 4$ different RTMs provides an indication of the maximum uncertainties resulting from the various numerical approximations implemented in these models. Within the spectral band $k$ of a radiometer, this range is expressed with the following equation

$$\overset{\leftrightarrow}{\Re}_k = \max_{\forall m \in M} \bar{B}_k(m) - \min_{\forall m \in M} \bar{B}_k(m). \qquad (4)$$

The values $\overset{\leftrightarrow}{\Re}_k$ have been estimated for all the spectral bands listed in Table 1. It provides useful information to RTM end-users concerning the uncertainties of vicarious calibration results due to the model numerical approximation.

## 6. Results

### 6.1. Aqua/Modis Results

The AQUA/MODIS radiometer has been in orbit for more than 20 years and provides observations of high radiometric stability [46]. As can be seen in Figure 2, the range of the mean bias $\overset{\leftrightarrow}{\Re}_k$ lays typically within 1–2% with the exception of band B07 centred around 2.1 μm, which exhibits a value as large as 3%. Molecular absorption in that spectral interval is particularly complex as several gas species are absorbing, as can be seen in Table 2. The approximation of molecular absorption differs according to the RTM as explained in Section 3. All models assume the same standard atmospheric profile in which the total column water vapor and ozone are rescaled using ERA-5 data from ECMWF as explained in Section 2.

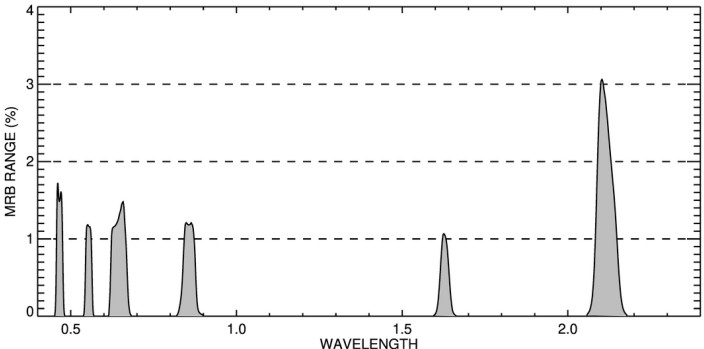

**Figure 2.** Plot of the relative range values expressed in percent calculated with Equation (4) for the processed MODIS bands listed in Table 1. The spectral location of the MODIS bands (shaded gray) is shown by their respective sensor spectral response (SSR), the magnitude of which is equal to the range value.

The best agreement between the four RTMs occurs in band B06 centred around 1.6 μm where the range of the estimated MRB is about 1%. In the visible spectral region, the range $\overset{\bullet\bullet}{\Re}_k$ takes its maximum value in blue band B03 with a mean relative difference between the RTMs of about 1.8%.

### 6.2. Envisat/Aatsr Results

The AATSR (Advanced Along Track Scanning Radiometer) radiometer of the ESA (European Space Agency) was on-board the Envisat platform which flew from 2002 up to 2012. It has now been replaced by SLSTR (Sea and Land Surface Temperature Radiometer) flying on-board the Sentinel-3 platform. AATSR has been however preferred to SLSTR, whose solar reflective band suffered from some radiometric calibration issues during the first years of operation.

All spectral bands show range values $\overset{\bullet\bullet}{\Re}_k$ lower than 1% except for band B4 centred around 1.6 μm, which exhibits a value exceeding 3% (Figure 3). This result significantly differs from the corresponding MODIS band B06 located in the same spectral region. However, theire exact shape and location are slightly different. The impact of these differences will be further analysed in Section 6.5.

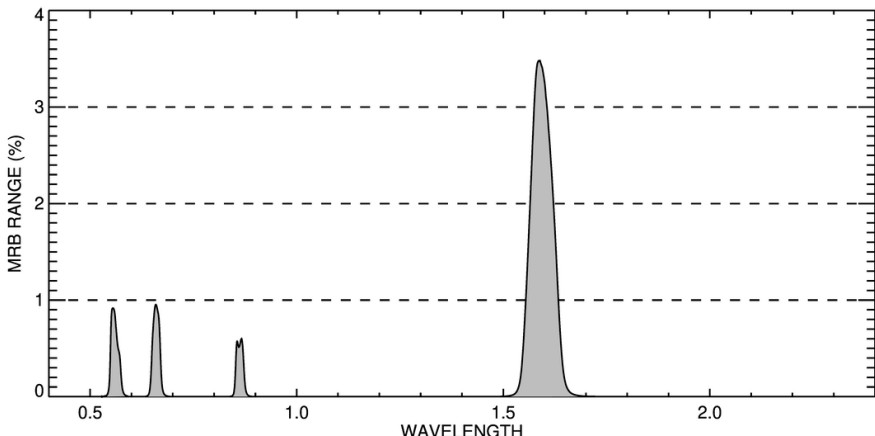

**Figure 3.** Same as Figure 2 but for the Envisat/AATSR instrument.

### 6.3. Landsat-8/Oli and Sentinel-2A/MSI Results

The Operational Land Imager (OLI) radiometer, like the ESA MultiSpectral Instrument (MSI), is a high-resolution instrument operated by United States Geological Survey (USGS). Both instruments have pretty similar spectral bands except for a few details. Results for these two radiometers are shown in Figures 4 and 5. The magnitude of $\overset{\bullet\bullet}{\Re}_k$ is very similar for both radiometers with the exception of MSI band 8 centred around 0.82 μm but with a spectral width of 105 nm. This band is severely affected by water vapor absorption, leading to a typical molecular transmittance of about 0.94 as opposed to 0.99 in MSI band 8A centred around 0.86 μm and a width of only 21 nm. Consequently, the range $\overset{\bullet\bullet}{\Re}_k$ in MSI band 8A or OLI band B5 is much smaller than 1% but exceeds 3% in MSI band 8. These results show that a reduction in radiometric noise resulting from an increase of the sensor bandwidth might come at a price of a higher inaccuracy in the simulations. The bands B6 (OLI) and B11 (MSI) centred around 1.6 μm have lower values than AATSR but higher values than MODIS. This conclusion does not hold for the 2.2 μm spectral region where these two radiometers exhibit lower $\overset{\bullet\bullet}{\Re}_k$ values than the corresponding MODIS band.

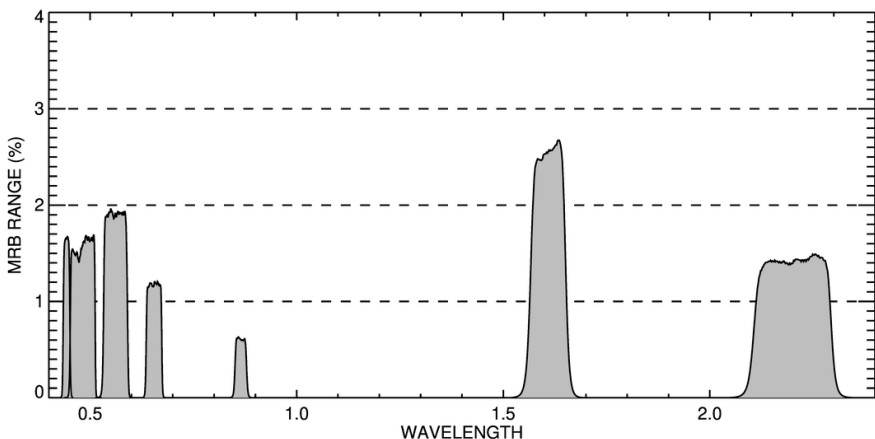

**Figure 4.** Same as Figure 2 but for the Landsat-8/OLI instrument.

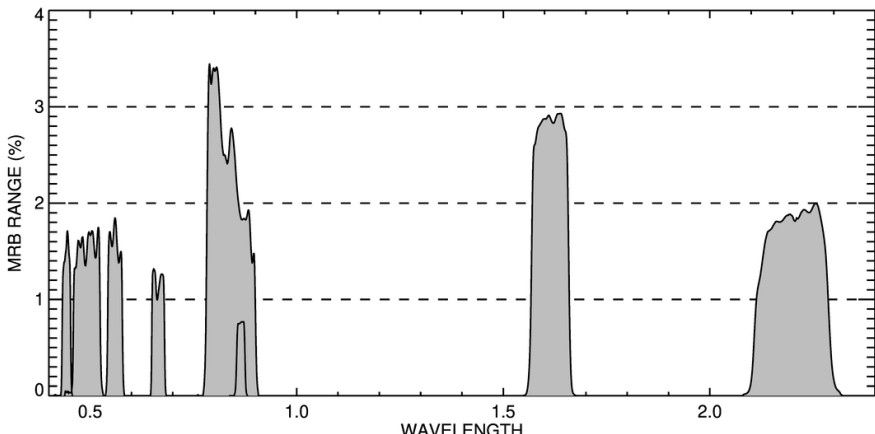

**Figure 5.** Same as Figure 2 but for the Sentinel-2A/MSI instrument.

### 6.4. Envisat/Meris and Sentinel-3A/OLCI Results

Medium Resolution Imaging Spectrometer (MERIS) and Ocean and Land Colour Instrument (OLCI) ESA radiometers are dedicated to ocean colour retrieval. Both instruments have narrow spectral bands, which is typically around 10 nm. The magnitude of $\overleftrightarrow{\Re}_k$ is very similar for both radiometers (Figures 6 and 7) exhibiting values as low as 0.6% in some spectral bands. For these bands and when the molecular transmittance is close to 0.99, RTMs typically agree within less than 1%. However, at short wavelengths, the range exceeds 1%, revealing potential numerical issues when scattering processes are very important, as explained in Section 6.5.

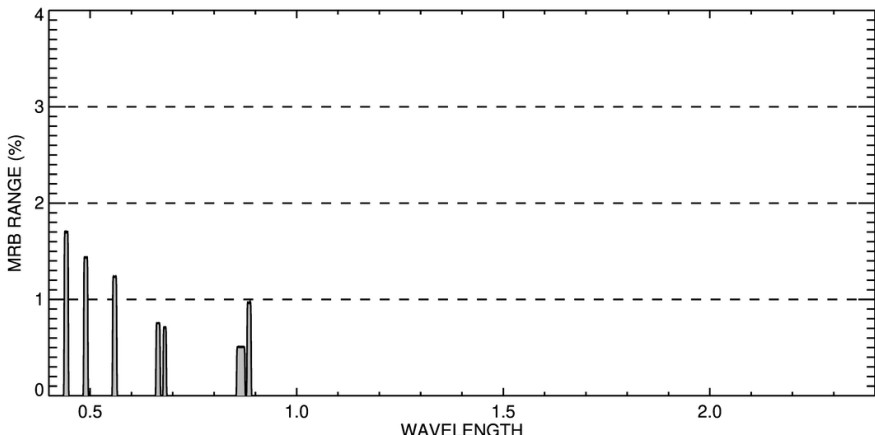

**Figure 6.** Same as Figure 2 but for the Envisat/MERIS instrument.

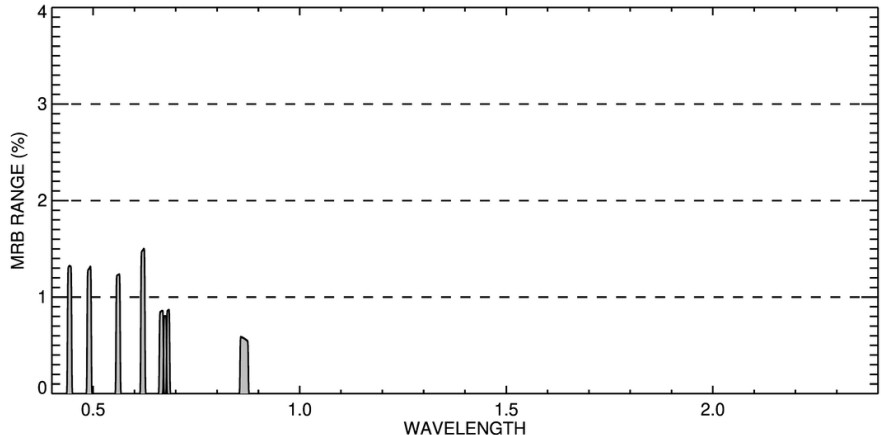

**Figure 7.** Same as Figure 2 but for the Sentinel-3A/OLCI instrument.

*6.5. Spectral Region Analysis*

In this section, the analysis of the $\overset{\leftrightarrow}{\Re}_k$ values is performed for the six spectral regions listed in Table 1 comparing the results for different radiometers. Figure 8 shows the results in the blue spectral domain. Radiative processes in this spectral regions are largely dominated by the scattering of light by molecules (Rayleigh) and aerosols. As can be seen in Figure 8, molecular absorption is minimal; the total gaseous transmittance exceeds 0.99 in most parts of the spectral region. Molecular atmosphere transmittance spectra were computed using the RADIS (RADIative Solver) [47] line-by-line code [48,49] using the HITRAN2020 molecular spectroscopic database [50] and partition function data from Gamache et al. [51]. Consequently, the $\overset{\leftrightarrow}{\Re}_k$ values are very similar, ranging from 1.3% to 1.7% , irrespectively of the SSR shape, width or location. This result is in line with values reported elsewhere [3]. It is important to stress here that all RTMs are operated in scalar mode; no polarisation effects are accounted for. The nature of these differences is due to the parametrisation of Rayleigh scattering, the numerical method to solve the multiple scattering, the number of quadrature points, and its coupling with a non-Lambertian surface.

Figure 9 shows the results in the green spectral region. The $\overset{\leftrightarrow}{\Re}_k$ values lay between 1% and 2%. Molecular absorption in this region is dominated by photodissociation-induced absorption by ozone molecules in the so-called Chappuis bands. A first analysis shows that wider bands exhibit larger values than narrower ones. Following this reasoning, MERIS or OLCI red band which have a bandwidth of about 10 nm should exhibit the smallest $\overset{\leftrightarrow}{\Re}_k$ values. However, each RTM relies on a different spectral resolution to solve the radiative transfer equation, potentially impacting the simulated values. 6SV has the lowest spectral

resolution at 2.5 nm, which is a value that cannot be customised. It might have an impact on the simulation of very narrow spectral bands. For the MERIS and OLCI bands, the $\overset{\leftrightarrow}{\Re}_k$ accounting only for RTMOM, libRadtran and ARTDECO is about 0.2%. In that specific case, the spectral resolution of 6SV is responsible for the $\overset{\leftrightarrow}{\Re}_k$ value to be close to 1%, although this model simulates multiple scattering very accurately. Its spectral resolution is, however, a limiting factor for very narrow spectral bands.

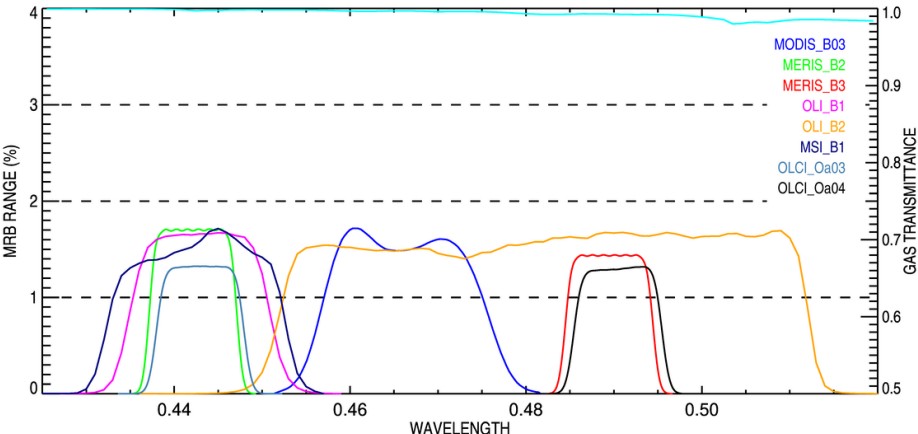

**Figure 8.** Plot of the MRB range values expressed in percent calculated with Equation (4) in the blue spectral region. The spectral location of the displayed radiometer bands is shown by their respective SSR, the magnitude of which is equal to the range. The right Y axis represents the total molecular transmittance displayed with the solid cyan line.

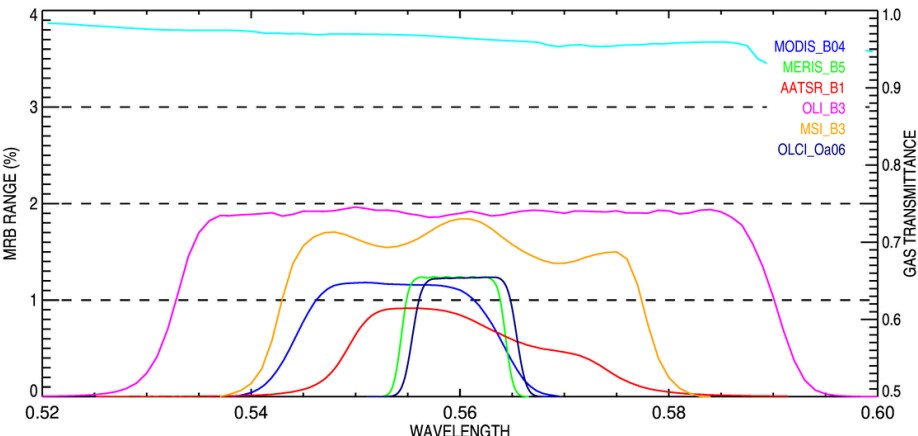

**Figure 9.** Same as Figure 8 but in the green spectral region.

The red spectral region (Figure 10) is subject to less molecular absorption than the green spectral region. Consequently, $\overset{\leftrightarrow}{\Re}_k$ hardly exceeds 1% to the exception of the OLCI band Oa07, which is affected by ozone absorption, leading to a value close to 1.5%. The average transmittance in that band is about 96.47% and is almost exclusively due to ozone.

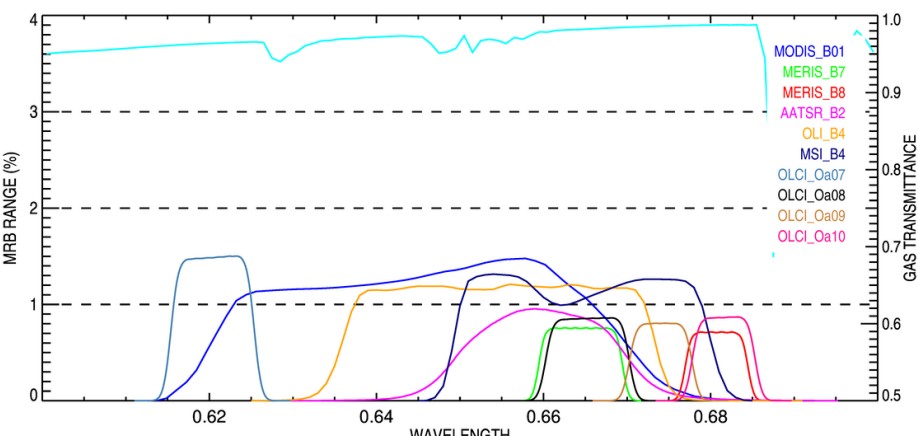

**Figure 10.** Same as Figure 8 but in the red spectral region.

The NIR spectral region (Figure 11) is also characterised by low $\overleftrightarrow{\Re}_k$ values below 1% for the narrow bands located around the 0.86 µm window. MSI band 8 is a noticeable exception in that spectral region. Important water vapor absorption lines are present around 0.8 µm, leading to a $\overleftrightarrow{\Re}_k$ value exceeding 3%. This value is reduced to 1.2%, considering the range between libRadtran, RTMOM and ARTDECO only. This decrease results from the limited capability of the 6SV model to deal with molecular absorption.

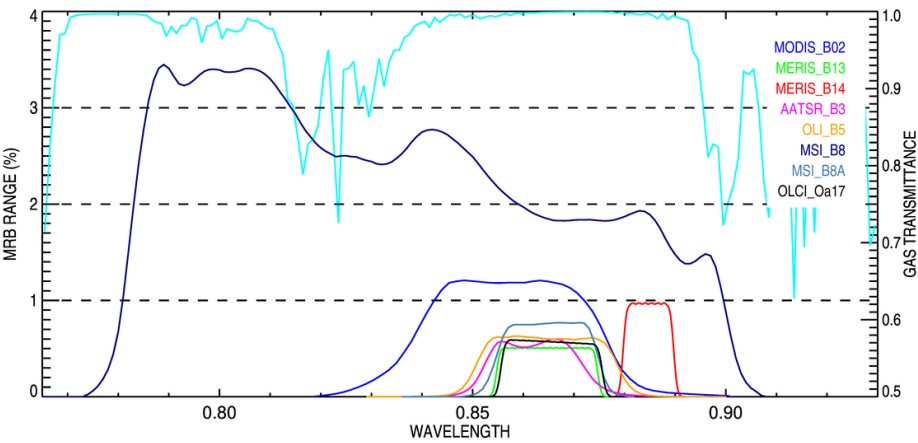

**Figure 11.** Same as Figure 8 but in the NIR spectral region.

The shortwave infrared region (SWIR) contains two atmospheric windows centred respectively around 1.6 µm (Figure 12) and 2.2 µm (Figure 13). These regions are subject to complex absorption processes by several different molecules such as carbon dioxide ($CO_2$), methane ($CH_4$) and water vapor ($H_2O$). Absorption by nitrous oxide ($N_2O$) occurs only in the SWIR2 spectral region. In the SWIR1 region, $\overleftrightarrow{\Re}_k$ is typically in the 2–3% range. MODIS band B06 is a noticeable exception thanks to its reduced width and optimal location. In the SWIR2 region, the OLI and MSI bands have very similar performance. MODIS band B07, centred at a slightly shorter wavelength, has more significant $\overleftrightarrow{\Re}_k$ value. The significant differences between the various bands can be explained by the values of the molecular transmittance displayed in Table 2.

**Table 2.** Molecular atmosphere's weighted average transmittance in percent for the spectral bands in the SWIR1 and SWIR2 regions listed in Table 1. The weighting function is the corresponding sensor spectral response. Transmittance values are given for the mid-latitude summer vertical profile. The last column corresponds to the total average transmittance. (1) Original $CO_2$ and $CH_4$ concentration (Conc.) values after [10]. (2) Current concentration values.

| | Band | $H_2O$ | $O_3$ | \multicolumn{2}{c}{Molecules $CO_2$} | $N_2O$ | \multicolumn{2}{c}{$CH_4$} | \multicolumn{2}{c}{Total} |
| Conc. | | 14.39 | 345.75 | 330[1] | 420[2] | 320 | 1700[1] | 1900[2] | (1) | (2) |
| Units | | kg m$^{-2}$ | DU | \multicolumn{2}{c}{ppmv} | ppbv | \multicolumn{2}{c}{ppbv} | | |
|---|---|---|---|---|---|---|---|---|---|---|
| MODIS | B06 | 99.6 | 100.0 | 99.4 | 99.3 | 100.0 | 99.5 | 99.5 | 98.6 | 98.4 |
| OLI | B6 | 99.4 | 100.0 | 98.3 | 98.0 | 100.0 | 99.7 | 99.7 | 97.5 | 97.1 |
| MSI | B11 | 99.4 | 100.0 | 98.4 | 98.0 | 100.0 | 99.6 | 99.6 | 97.4 | 97.0 |
| AATSR | B4 | 99.3 | 100.0 | 98.0 | 97.6 | 100.0 | 99.9 | 99.9 | 97.3 | 96.8 |
| MODIS | B07 | 95.6 | 100.0 | 98.5 | 98.1 | 99.8 | 99.9 | 99.9 | 93.9 | 93.6 |
| OLI | B7 | 96.4 | 100.0 | 99.9 | 99.8 | 99.8 | 97.7 | 97.5 | 93.9 | 93.6 |
| MSI | B12 | 96.4 | 100.0 | 99.9 | 99.9 | 99.8 | 97.8 | 97.6 | 94.0 | 93.8 |

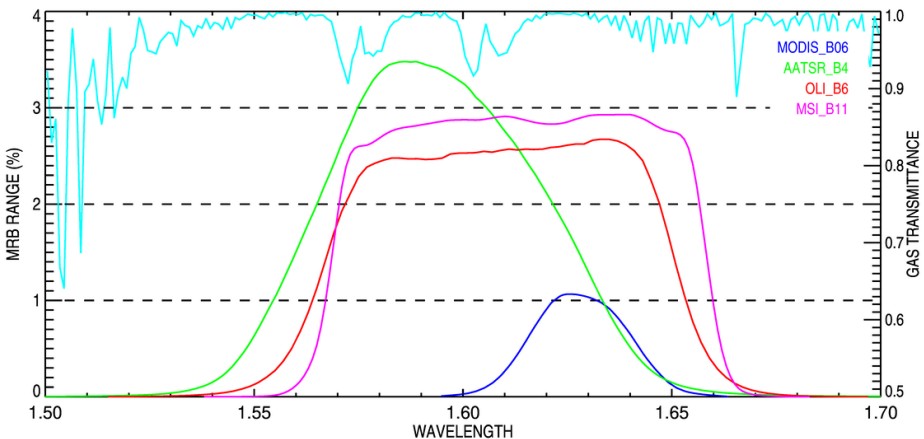

**Figure 12.** Same as Figure 8 but in the 1.6 μm spectral region.

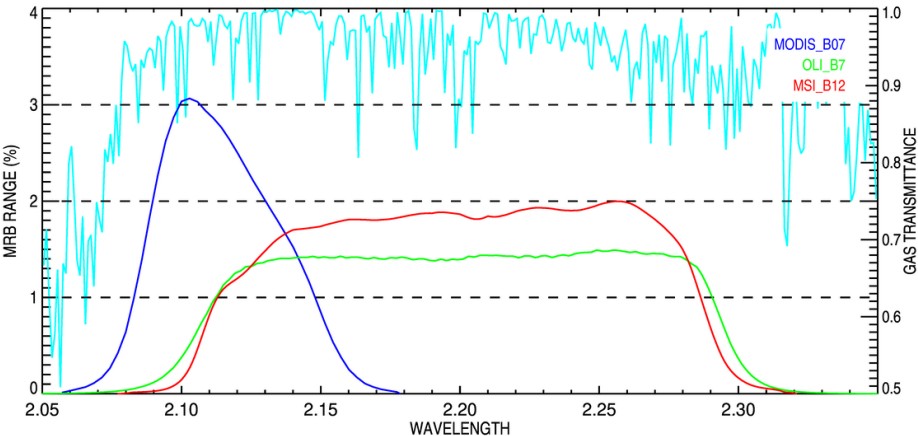

**Figure 13.** Same as Figure 8 but for the 2.2 μm spectral region.

## 7. Discussion and Conclusions

Four different radiative transfer models have been compared in actual application conditions, namely vicarious calibration over bright desert PICS. This study thus represents a benchmark of these models in typical end-users usage. The metrics defined to compare these models is based on the MRB between observation and simulation, concretely the

range of MRB values between the four RTMs. This metric is different from the approach proposed by the Radiation Transfer Model Intercomparison (RAMI) initiative [17], which is designed to identify models that show large discrepancies with respect to the average simulated results. This research focuses on the characterisation of the difference between the models, irrespectively of their actual performance. It is primarily addressed to the end-users of these models interested in the numerical uncertainty estimation of simulated satellite observations.

This study reveals that a vicarious calibration method relying on simulated satellite observations might be impacted by the choice of the radiative transfer model from 0.5% up to 3.5% according to the spectral bandwidth and location. The largest differences primarily occur in spectral regions affected by the molecular absorption of different species. The exact location and width of the sensor spectral response might therefore have an important impact on the simulation accuracy. The width of the bands also contributes to model discrepancies, as large bands are more likely to contain molecular absorption lines significantly affecting the magnitude of the reflected radiance. In spectral regions where the molecular absorption is very low, i.e., the corresponding transmittance is larger than 0.99 like in the NIR domain (Figure 11), the model discrepancy is in the range 0.6–0.8%. It is currently the best agreement between these models when operating in their standard configuration. The blue spectral region is a noticeable exception where model discrepancies usually exhibit values around 1.7% in scalar mode.

The analysis of each radiometer individually (Figures 2–7) does not reveal features specific to a given sensor. The analysis of the different spectral regions (Figures 8–13) uncovers the decisive role of the sensor spectral response location with respect to the molecular absorption lines to explain the model differences. The 6SV model should be avoided in cases of molecular absorption but performs well in spectral regions where radiative processes are dominated by scattering. Conversely, the ARTDECO model is shipped with a comprehensive CKD datasets providing accurate results but at prohibitive computing time.

New missions such as CLARREO (Climate Absolute Radiance and Refractivity Observatory) [52] or TRUTHS (Traceable Radiometry Underpinning Terrestrial- and Helio-Studies) [53] will deliver observations with unprecedented radiometric accuracy with respect to current mission. It targets the delivery of radiances with an accuracy better 1%, which is a challenge for RTM developers. Reaching an agreement between the models irrespectively of the simulated spectral bands will require additional efforts in their usage and possible improvements:

- In spectral regions dominated by scattering and in the presence of large particles, models based on adding–doubling or discrete ordinate should be operated with a large number of quadrature points. A numerical method such as Monte Carlo ray tracing should be favoured in the presence of large particles.
- In spectral regions where molecular absorption is dominant, i.e., with a corresponding transmittance lower than 0.98, it is essential that all active molecules are accounted for. There is, however, an intrinsic limitation to band-integrated transmittance methods of about 1% in the worst cases [16]
- There is also an intrinsic limitation to the 1D approximation when topography effects have to be considered [23]. The systematic usage of 3D RTM might be needed to deliver simulated radiance compatible with missions such as CLARREO or TRUTHS over complex terrains.
- Finally, it is essential to permanently benchmark RTMs using scenarios mimicking actual usage. The latest RAMI initiative [54], dedicated to the benchmarking of coupled surface-atmosphere radiative transfer models, is particularly relevant in this context. This type of benchmarking is based on scenarios of gradual complexity which will allow one to characterise and understand in great detail the discrepancies between the models.

**Author Contributions:** Conceptualisation, Y.G.; methodology, Y.G.; software, Y.G. and Y.N.; validation, Y.G., Y.N. and V.L.; formal analysis, Y.G.; writing—original draft preparation, Y.G.; writing—review and editing, Y.N. and V.L. All authors have read and agreed to the published version of the manuscript.

**Funding:** This research received no external funding.

**Institutional Review Board Statement:** Not applicable.

**Informed Consent Statement:** Not applicable.

**Acknowledgments:** The authors thanks the Laboratoire d'Optique Atmosphérique of the Lille University for their support for the use of the ARTDECO model.

**Conflicts of Interest:** The authors declare no conflict of interest.

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
