# Peer review of "Radiative Transfer Model Comparison with Satellite Observations over CEOS Calibration Site Libya-4"

_atmosphere, doi:10.3390/atmos13111759_

Round 1

Reviewer 1 Report

RTM is the basic tool when the satellite sensor perform the vicarious calibration, especially in the using of PICs method.  This paper estimated the uncertainty of the simulated radiance at the top of atmosphere for the 6 sensor over Libya-4 by using 4 RTM. The results and conclusions are very interesting and helpful for the end-users to perform the vicarious calibration for the satellite sensors.

The paper is suitable to published in this journal in general while I have a few comments for discussion.

1. Page 2, Line 49: "complexifies" should be "simplifies". I am not sure if I understood correctly.

2. The six instruments have been used in this paper. The calibration accuracy of these instruments is in the different level. Such difference will affact the O-B (observation minus simulation) therefore will affect the value of  the RMB range. Did such difference has been considered in the paper? I suggested to discuss the effect of calibration accuracy on the conclusion of this paper.

3. In the paper, from Fig 2 to Fig 7, what is the value of MRB. Did the absolute value of MRB affects the MRB range? Is it possible to show the MRB and MRB range together in the plot? Current plot is not a good way to show the author's content.

4. Only Libya-4 case was calculated in the paper. It means the BRF in a certain high value of reflectivity for the solar reflection band. For the low value range of reflectivity, such as over the forest or the ocean, what is the conclusions? Although this paper discusses the results over the Libya-4, if the errors over the low reflectivity can be discussed in the conclusion, this paper will more interested for the end-users.

Author Response

Comments and Suggestions for Authors

RTM is the basic tool when the satellite sensor perform the vicarious calibration, especially in the using of PICs method.  This paper estimated the uncertainty of the simulated radiance at the top of atmosphere for the 6 sensor over Libya-4 by using 4 RTM. The results and conclusions are very interesting and helpful for the end-users to perform the vicarious calibration for the satellite sensors.

The paper is suitable to published in this journal in general while I have a few comments for discussion.

  1. Page 2, Line 49: "complexifies" should be "simplifies". I am not sure if I understood correctly.

Response 1:

In the following sentences (lines 50 to 62), it is explained why simulating a spectral band is much more complex than the simulation of monochromatic radiation:

Simulations of radiances acquired by a radiometer such as MODIS (Moderate Resolution Imaging Spectroradiometer)  require to integrate the simulated radiative field over the observed spectral domain.  The most obvious way to perform this spectral integral consists in solving the radiative transfer equation for each wavelength of a spectral mesh dense enough to resolve the line structure of the absorption spectrum of the atmosphere.  …

  1. The six instruments have been used in this paper. The calibration accuracy of these instruments is in the different level. Such difference will affact the O-B (observation minus simulation) therefore will affect the value of  the RMB range. Did such difference has been considered in the paper? I suggested to discuss the effect of calibration accuracy on the conclusion of this paper.

Response 2:

The reviewer raised a very good point. Equation (3) defines the mean relative bias (MRB) between observation and the simulated calibration reference which is used for radiometric calibration verification. It is however not the metric which is used here but the one defined by Equation 4. This metric measures the ranges of MRB between the 4 RTMs for a given spectral band. This range is therefore independent from the actual radiometric calibration of the radiometer. This statement would not be true in case an instrument would exhibit very large structured uncertainties which is not the case for the selected instrument. Random uncertainties are close to zero as many pixels are averaged over the selected PICS. The following sentence has been added in Section 4 around line 219  

These radiometers have a very low radiometric noise.

  1. In the paper, from Fig 2 to Fig 7, what is the value of MRB. Did the absolute value of MRB affects the MRB range? Is it possible to show the MRB and MRB range together in the plot? Current plot is not a good way to show the author's content.

Response 3:

This paper is not about the analysis of MRB values. This work has been performed in reference [27]. It focuses on the MRB difference between the 4 RTMs as given by Equation (4). We invite reviewer 1 to read reference [27] for the MRB analysis.

  1. Only Libya-4 case was calculated in the paper. It means the BRF in a certain high value of reflectivity for the solar reflection band. For the low value range of reflectivity, such as over the forest or the ocean, what is the conclusions? Although this paper discusses the results over the Libya-4, if the errors over the low reflectivity can be discussed in the conclusion, this paper will more interested for the end-users.

Response 4:

This comment is very relevant. The objective of this paper is to illustrate the difference between RTMs when used for a specific application. This first analysis shows that these differences vary with the spectral region and shape of the sensor spectral response. It is indeed expected that many other factors will impact these difference such the surface type, atmospheric composition, … The RAMI4ATM initiative (https://rami-benchmark.jrc.ec.europa.eu/_www/RAMI4ATM.php) has been specifically set up to address a large variety of cases. 

Reviewer 2 Report

The manuscript compares four different 1D radiative transfer models corresponding to the simulation of satellite observations with observations acquired over the pseudo-invariant calibration site Libya-4. Differences between 0.5 and 3.5% are found between models, varying by the spectral region and the shape of the sensor spectral response. The manuscript is organized well with sound proofs and comprehensive discussions. I just have a couple of technical questions and suggest minor revisions before publication.

1.      Line 106: what’s the threshold of “clear-sky”? Any cloud radiance fraction criteria applied?

2.      I’m confused by the combination of satellite and different radiative transfer models, as stated in Line 211. Why do you need to rely on the simulation of actual satellite observations? If you want to compare different radiative transfer models with the measurements, why not apply these four models with one same satellite? Please elaborate more.

3.      Line 330: what’s the merit of the metrics proposed in this study compared to the matric proposed by RAMI initiative?

Author Response

Comments and Suggestions for Authors

The manuscript compares four different 1D radiative transfer models corresponding to the simulation of satellite observations with observations acquired over the pseudo-invariant calibration site Libya-4. Differences between 0.5 and 3.5% are found between models, varying by the spectral region and the shape of the sensor spectral response. The manuscript is organized well with sound proofs and comprehensive discussions. I just have a couple of technical questions and suggest minor revisions before publication.

  1. Line 106: what’s the threshold of “clear-sky”? Any cloud radiance fraction criteria applied?

Response 1

The cloud mask available from the various sensors are used to identify the presence of cloud. All pixels should be cloud free. The following sentence has been modified in Section 4

Only clear-sky satellite observations are considered.

reads now

Only clear-sky satellite observations are considered, meaning that all pixels over the PICS must be flagged as cloud-free.

  1. I’m confused by the combination of satellite and different radiative transfer models, as stated in Line 211. Why do you need to rely on the simulation of actual satellite observations? If you want to compare different radiative transfer models with the measurements, why not apply these four models with one same satellite? Please elaborate more.

Response 2

All RTMs are compared to all different radiometers. There are no combination of satellites and RTMs. We rely on the comparison of satellite observations to compare these models in actual usage conditions as stressed in the abstract and in line 76. The objective was not to use artificial benchmark scenarios that do not represent practical usage of these model.

  1. Line 330: what’s the merit of the metrics proposed in this study compared to the matric proposed by RAMI initiative?

Response 3

The purpose of RAMI is to establish a consensus on the most likely solution corresponding to the described scene and to rank the model according to this solution. The metric used in this study aims at illustrating the difference between RTMs when the same satellite observations are simulated.